# PARP1 exhibits enhanced association and catalytic efficiency with γH2A.X-nucleosome

Deepti Sharma [1], Louis De Falco [1], Sivaraman Padavattan[1,3], Chang Rao[1], Susana Geifman-Shochat[1,2], Chuan-Fa Liu [1,2] & Curt A. Davey [1,2]*

The poly(ADP-ribose) polymerase, PARP1, plays a key role in maintaining genomic integrity by detecting DNA damage and mediating repair. γH2A.X is the primary histone marker for DNA double-strand breaks and PARP1 localizes to H2A.X-enriched chromatin damage sites, but the basis for this association is not clear. We characterize the kinetics of PARP1 binding to a variety of nucleosomes harbouring DNA double-strand breaks, which reveal that PARP1 associates faster with (γ)H2A.X- versus H2A-nucleosomes, resulting in a higher affinity for the former, which is maximal for γH2A.X-nucleosome that is also the activator eliciting the greatest poly-ADP-ribosylation catalytic efficiency. The enhanced activities with γH2A.X-nucleosome coincide with increased accessibility of the DNA termini resulting from the H2A.X-Ser139 phosphorylation. Indeed, H2A- and (γ)H2A.X-nucleosomes have distinct stability characteristics, which are rationalized by mutational analysis and (γ)H2A.X-nucleosome core crystal structures. This suggests that the γH2A.X epigenetic marker directly facilitates DNA repair by stabilizing PARP1 association and promoting catalysis.

[1] School of Biological Sciences, Nanyang Technological University, 60 Nanyang Drive, Singapore 637551, Singapore. [2] NTU Institute of Structural Biology, Nanyang Technological University, 59 Nanyang Drive, Singapore 636921, Singapore. [3] Present address: Department of Biophysics, National Institute of Mental Health and Neurosciences, Bangalore 560029, India. *email: davey@ntu.edu.sg

PARP1 (poly(ADP-ribose) polymerase 1) is a multipartite polymerase that assumes a fundamental role in maintaining genomic stability by acting as a regulator of chromatin composition and structure, transcription and DNA damage repair[1–3]. The enzyme specifically recognizes DNA single-strand break (SSB) and DNA double-strand break (DSB), resulting in activation of the poly-ADP-ribosylation (PARylation) activity towards itself (auto-modification/PARylation) and other protein substrates, including core and linker histones. The PARylation of PARP1 and other nuclear factors evokes a cascade molecular response that leads to recruitment of additional DNA repair factors and chromatin remodelling machinery to the site of DNA damage. Indeed, the sensitivity of BRCA1/BRCA2-deficient tumour cells towards PARP inhibition has rendered PARP1 an important cancer drug target[4].

PARP1 consists of six domains, four of which are responsible for recognition of DNA lesions, including three zinc-finger motifs in the N-terminal half and a WGR domain that is linked to the zinc-fingers via a BRCT domain[1] (Fig. 1a). The BRCT domain and adjacent C-terminal linker comprise the sites of enzymatic auto-modification accompanying PARP1 activation. The catalytic activity resides at the C-terminus of PARP1, which consists of an ADP-ribosyl transferase (ART) fold and a regulatory helical subdomain (HD). In the resting state, the folded conformation of the HD acts to block $NAD^+$ access to the ART active site[5]. Upon association with a SSB or DSB, unfolding of the HD is allosterically triggered, which in turn allows unrestricted access of $NAD^+$ to the active site.

A study characterizing PARP1 binding modes showed that the enzyme interacts with higher affinity towards certain nucleosomal compared to naked DNA DSB constructs and that this selectivity requires both the N- and C-terminal halves of PARP1[6]. Moreover, the preferential binding of nucleosomal targets occurs only when linker DNA is present at both termini, and since PARP1 binds to the nucleosome with 1:1 stoichiometry, this suggests that the polymerase interacts with the two linker DNA arms simultaneously. This raised the interesting possibility of a more elaborate interplay between chromatin and PARP1 in DNA damage detection and potentially downstream events.

Although PARP1 is highly prevalent in vivo, with a stoichiometry of roughly one molecule for every 30 nucleosomes in human cells[7], it is not evenly distributed throughout chromatin. Instead, PARP1 tends to be localized in the vicinity of promotor regions of particular genes[8,9], with specific enrichment at H2A.X and H2A.Z nucleosomes in mammalian mitotic chromatin[10]. Moreover, the positioning of H2A.X nucleosomes has been shown to coincide with endogenous DSB hotspots[11], suggesting that damage-prone regions in chromatin could be primed for repair. This would be consistent with the function of γH2A.X—the Ser139 phosphorylation product of H2A.X—as the primary epigenetic mark for delineation of DSBs[12].

Upon DSB formation, PARP1 binding to the lesion is among the first molecular responses to take place[13]. DSB recognition by and activation of PARP1 is proceeded by actual phosphorylation of H2A.X[13], which ultimately culminates in γH2A.X foci formation to induce a cascade repair response[14]. Given the preferential association of PARP1 to nucleosomal DSBs in vitro[6,15] and with H2A.X/γH2A.X chromatin sites in the cell[10,13,16,17], we wondered whether the variant histone and post-translational modification could directly influence PARP1 DSB binding and activity. Indeed, studies have suggested that H2A.X/γH2A.X chromatin has reduced stability relative to canonical H2A chromatin[17–19].

Here, we produced γH2A.X using a hybrid chemical approach and assembled nucleosomes with either H2A, H2A.X, γH2A.X or mutants of H2A or H2A.X as the incorporated H2A variant.

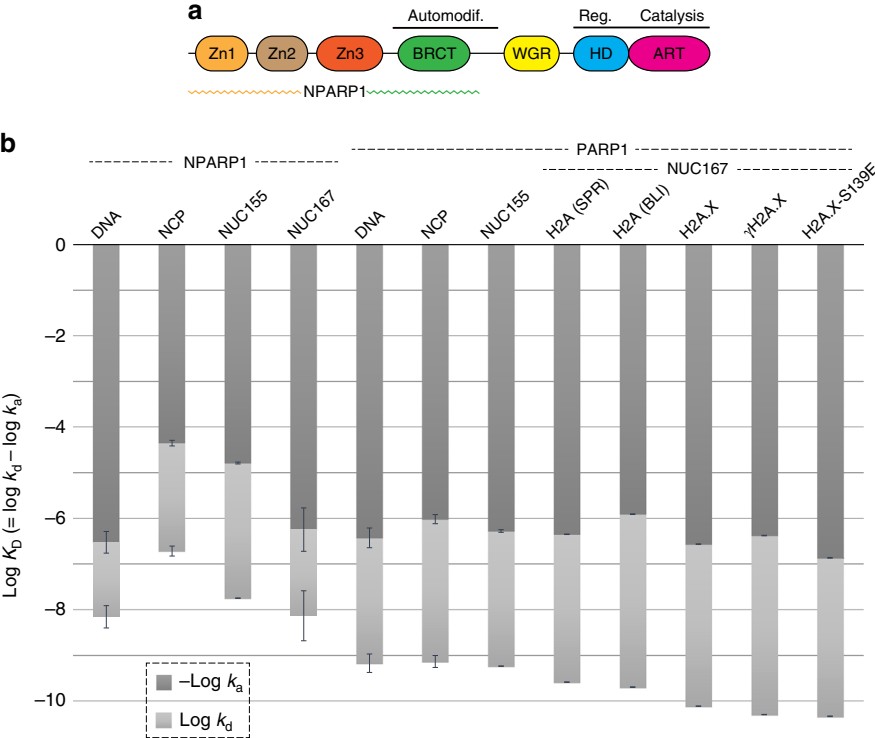

**Fig. 1 PARP1 association with DSB activators. a** Domain structure of the 1024 amino acid PARP1 and corresponding N-terminal half truncation construct (NPARP1). **b** Kinetic parameters for NPARP1 and PARP1 binding to naked DNA, NCP and nucleosomal substrates measured by SPR (eight samples, left; mean ± s.d., $n = 2$ independent experiments) and BLI (four samples, right; mean ± s.d., $n = 3, 2, 3, 2$ independent experiments). Source data are provided as a Source Data file.

Nucleosome stability/dynamics investigations and crystal-lographic structure characterization were carried out in combination with surface plasmon resonance (SPR)- and bio-layer interferometry (BLI)-based techniques to measure the kinetics of PARP1 association with and dissociation from the DSB constructs (represented by the blunt-end DNA termini; *Homo sapiens* histone and PARP proteins). Additionally, enzymatic assays were performed to shed light on the catalytic parameters of PARylation. This reveals that PARP1 recognition of and activation by DSBs is modulated by histone variant composition and post-translational modification status.

## Results

**PARP1 binds preferentially to H2A.X/γH2A.X nucleosomes**. In seeking a label-free method for monitoring nucleosome-protein factor interactions in real time, we established approaches with SPR and BLI to measure the kinetics of PARP1 and NPARP1 (N-terminal half of PARP1, lacking the WGR and catalytic domains; Fig. 1a) association with different DNA damage constructs. The DSB constructs include a 145 bp nucleosome core particle (NCP), a nucleosome with a single double-helical turn linker DNA at one terminus (NUC155), a nucleosome with a single double-helical turn linker DNA at both termini (NUC167) and the corresponding 167 bp DNA.

Consistent with the study of Clark et al.[6], PARP1 has the highest affinity for nucleosomal constructs having linker DNA extensions at both termini (Fig. 1b; Supplementary Figs. 1 and 2; Supplementary Table 1). Moreover, the selectivity towards such nucleosomes (NUC167) over naked DNA requires domains in the C-terminal half as this preference is not displayed by NPARP1. The C-terminal half of PARP1 also provides an added driving force for binding to any DSB constructs[6]. We see here that the increased affinities associated with the full-length polymerase are primarily the result of decreased PARP1 dissociation rates ($k_d$) for either NUC167 or DNA, but increased PARP1 association rates ($k_a$) towards NCP145 and NUC155. The affinities observed for PARP1 binding are higher than found previously based on a HI-FI FRET (fluorescence resonance energy transfer) methodology[6,15], with the lowest dissociation constant, $K_D = 246 \pm 3$ pM (SPR) or $191 \pm 3$ pM (BLI), seen with the current set of constructs for NUC167. The reduced affinity for DNA ($K_D = 638 \pm 20$ pM) is primarily a result of a faster PARP1 dissociation rate from the naked DNA.

In order to shed light on the potential influence of histone composition and modification status on PARP1 activity, in addition to 'canonical' NUC167, we measured PARP1 binding kinetics with either H2A.X, γH2A.X or H2A.X-S139E (mutant mimic of γH2A.X) in place of the H2A. These measurements show that PARP1 has a systematically higher affinity for the H2A.X-containing nucleosomes compared to the H2A nucleosome (Fig. 1b; Supplementary Fig. 3; Supplementary Table 1). This results primarily from an increased PARP1 association rate with the X variant nucleosomes. Compared to the H2A.X nucleosome ($K_D = 73.2 \pm 0.3$ pM), the γH2A.X and H2A.X-S139E nucleosomes have the higher affinity for PARP1, with $K_D = 47.8 \pm 0.1$ and $43.8 \pm 0.1$ pM, respectively. Nonetheless, we do not observe significant affinity of PARP1 for isolated histone complexes, including (dissociated) octamer, (H3-H4)$_2$ tetramer or N-H2B dimer, where N is either H2A, H2A.X or γH2A.X, in addition to γH2A.X ($K_D > 0.5$–10 μM).

**H2A/H2A.X/γH2A.X nucleosomes: very similar structures**. In order to shed light on the basis for the variant-specific differences in PARP1 nucleosome association, we sought to obtain well-diffracting crystals with NCP, which required screening of

different DNA constructs. The structures of H2A.X-NCP, γH2A.X-NCP, H2A.X-S139E-NCP and H2A-NCP were solved at 2.85, 2.75, 2.2 and 1.99 Å resolution, respectively (Supplementary Fig. 4 and Tables 1 and 2).

To obtain well-diffracting crystals of γH2A.X-NCP, we designed a cohesive-end DNA fragment (147s) consisting of a 143 bp core with 4-nucleotide 3′ overhangs at each terminus. This construct assembled into NCP crystallizes into pseudo-continuous chromatin fibres in the lattice, with the two DNA termini from a given NCP forming Watson–Crick base pairing with the complementary termini of two different NCPs (structure also solved for H2A-NCP assembled with this DNA fragment at 2.25 Å resolution; Fig. 2a and Table 2). By 'locking in' the entry/exit points of the nucleosome core, this (type of) construct may generally facilitate structural characterization of nucleosomal systems having diminished stability (elevated dynamics). Accordingly, the X variant structures were compared to respective H2A-containing NCPs, which shows that substitution with the H2A.X variant causes no gross conformational alterations within the histone octamer (Fig. 2b; r.m.s.d. from superimposing the histone fold domains of the octamer is ~0.33 Å when the assembled DNA fragment is the same and ~0.71 Å when it is different; see Methods).

The H2A.X variant primary structure differs mainly from that of H2A with respect to the C-terminal tail, after residue 120 (Supplementary Fig. 5). The 142-residue H2A.X has an additional 13 residues at the C-terminus, while most of residues 121–129 are different relative to H2A. Otherwise, there are only two amino acids that are different in the N-terminal tail region and two that differ (canonically) within the histone fold and extension domains. The conformation of the C-terminal tail of H2A/H2A.X is highly variable beyond residue 117, and in the X variant structures it is too disordered after residue ~124 to model (Fig. 2b). Therefore, the C-terminal tail of H2A.X, like that of H2A, is disordered in the nucleosome, at least in the absence of additional interacting factors. Likewise, the N-terminal tail in the X variant structures, as in H2A structures, is also variable/disordered.

**Table 1 Data collection and refinement statistics for NCP assembled with H2A.X, γH2A.X or H2A.X-S139E.**

|  | H2A.X | γH2A.X | H2A.X-S139E |
|---|---|---|---|
| **Data collection** |  |  |  |
| Space group | P2$_1$2$_1$2$_1$ | P2$_1$2$_1$2$_1$ | P2$_1$2$_1$2$_1$ |
| Cell dimensions |  |  |  |
| $a, b, c$ (Å) | 105.8, 109.9, 180.4 | 102.9, 109.4, 181.5 | 108.0, 109.5, 183.4 |
| $\alpha, \beta, \gamma$ (°) | 90, 90, 90 | 90, 90, 90 | 90, 90, 90 |
| Resolution (Å) | 2.85–93.9 (2.85–3.00)[a] | 2.75–93.7 (2.75–2.90)[a] | 2.20–94.0 (2.20–2.32)[a] |
| $R_{merge}$ (%) | 19.3 (121) | 19.3 (252) | 6.6 (113) |
| $R_{pim}$ (%) | 9.0 (65.5) | 5.9 (77.9) | 3.5 (73.5) |
| $I/\sigma I$ | 5.2 (1.1) | 8.4 (1.1) | 11.6 (1.0) |
| CC½ (%) | 99.5 (59.8) | 99.9 (45.7) | 99.5 (51.4) |
| Completeness (%) | 99.9 (99.8) | 100 (100) | 98.6 (91.2) |
| Redundancy | 5.5 (4.2) | 11.9 (11.3) | 5.4 (3.3) |
| **Refinement** |  |  |  |
| Resolution (Å) | 2.85–93.9 | 2.75–93.7 | 2.20–94.0 |
| No. of reflections | 47,274 | 52,860 | 106,763 |
| $R_{work}/R_{free}$ (%) | 26.5/32.0 | 22.0/27.6 | 23.2/27.4 |
| No. of atoms | 12,157 | 12,255 | 12,281 |
| Protein | 6174 | 6169 | 6233 |
| DNA | 5939 | 6029 | 5939 |
| Solvent | 44 | 57 | 109 |
| $B$-factors (Å$^2$) | 98 | 95 | 74 |
| Protein | 68 | 74 | 56 |
| DNA | 129 | 117 | 93 |
| Solvent | 61 | 81 | 61 |
| R.m.s. deviations |  |  |  |
| Bond lengths (Å) | 0.006 | 0.004 | 0.008 |
| Bond angles (°) | 1.428 | 1.210 | 1.482 |

[a]Single crystal data sets; values within parentheses are for the highest resolution shell

**Table 2 Data collection and refinement statistics for NCP assembled with H2A and either the same blunt-end 145 bp DNA fragment (NCP145) used for the H2A.X structure or the same sticky-end 147 bp DNA fragment (NCP147s) used for the γH2A.X structure.**

|  | NCP145 | NCP147s |
|---|---|---|
| Data collection |  |  |
| Space group | $P2_12_12_1$ | $P2_12_12_1$ |
| Cell dimensions |  |  |
| $a, b, c$ (Å) | 107.6, 109.7, 183.5 | 105.3, 109.7, 183.8 |
| $\alpha, \beta, \gamma$ (°) | 90, 90, 90 | 90, 90, 90 |
| Resolution (Å) | 1.99–76.8 | 2.25–94.2 |
|  | (1.99–2.10)[a] | (2.25–2.37)[a] |
| $R_{merge}$ (%) | 6.8 (212) | 9.5 (194) |
| $R_{pim}$ (%) | 2.1 (65.2) | 3.0 (76.0) |
| $I/\sigma I$ | 17.4 (1.3) | 12.2 (1.0) |
| CC½ (%) | 99.7 (52.6) | 99.7 (61.6) |
| Completeness (%) | 99.6 (97.9) | 99.7 (97.7) |
| Redundancy | 12.2 (11.6) | 11.5 (7.9) |
| Refinement |  |  |
| Resolution (Å) | 1.99–76.8 | 2.25–94.2 |
| No. of reflections | 145,312 | 99,086 |
| $R_{work}/R_{free}$ (%) | 23.6/26.3 | 23.9/29.2 |
| No. of atoms | 12,178 | 12,227 |
| Protein | 6,118 | 6,129 |
| DNA | 5,939 | 6,029 |
| Solvent | 121 | 69 |
| $B$-factors (Å$^2$) | 79 | 85 |
| Protein | 53 | 63 |
| DNA | 106 | 108 |
| Solvent | 44 | 66 |
| R.m.s. deviations |  |  |
| Bond lengths (Å) | 0.008 | 0.007 |
| Bond angles (°) | 1.474 | 1.496 |

[a]Single crystal data sets; values within parentheses are for the highest resolution shell

Beyond differences between the tails that could modulate dynamic properties, the two amino acid changes in the histone fold/extension domains could also have a direct influence by altering both the (H2A-H2B) dimer–(H3-H4) tetramer and dimer–dimer interfaces that stabilize the histone octamer (Fig. 3). The substitution of Gly (H2A.X) in place of Arg (H2A) at residue 99 within the docking domain eliminates a hydrogen bond between the Arg side chain and residue G94 of H4 (Fig. 3a). Nonetheless, by maintaining that hydrogen bonding position, the H2A-R99 guanidinium group is situated in immediate proximity with another guanidinium group, that of H4-R95, only 4.3 Å distant.

The other histone fold substitution entails His (H2A.X) in place of Asn (H2A) at residue 38. This alteration impacts the H2A loop 1 (L1) and H2B C-terminal portion of helix 3 (α3) region, which coincides with the sole motif of interaction between the two dimers (Fig. 3b). In the H2A-NCP structures, the H2A L1-H2A′ L1 contact is stabilized by side chain hydrogen bonding between H2A N38 and H2B′ H79. With the change to H38 in H2A.X, this residue is still capable of hydrogen bonding with H2B H79, but inspection of the X variant structures shows a substantial degree of variability in the side chain and even back bone conformations of the L1 region and H2B H79 (Fig. 3c). This suggests that at least part of the time—within the crystal solvent milieu at pH 6.0— H2A.X H38 and H2B H79 are both protonated (charged) and therefore incapable of interacting with each other favourably. Notably, H2A N38 can in principle hydrogen bond with H2B H79 in any protonation state. Although it is not conserved among the common H2A subtypes, the H2A.1 subtype

we employ here has Ser at position 40, which is an Ala residue in H2A.X (and other H2A subtypes; Fig. 3b and Supplementary Fig. 5). By eliminating a hydrogen bond within L1, the Ala substitution could impact the dynamics of this loop region.

**H2A/H2A.X/γH2A.X nucleosomes: distinct stability and dynamics.** Tyrosine fluorescence spectroscopy was employed to test the stability of the H2A.X-type nucleosomes with respect to salt-induced dissociation. The assay reveals a significantly lower salt stability for the H2A.X, γH2A.X and H2A.X-S139E nucleosomes relative to the H2A nucleosome (Fig. 4a). In contrast, high temperature-induced dissociation assays show the reverse trend, whereby the H2A.X and γH2A.X nucleosomes are thermally more stable than that of H2A (Fig. 4b).

In addition, we assessed the stability of two more mutants, wherein the non-tail residues that are canonically distinct between H2A and H2A.X were exchanged in the two variants, yielding H2A-N38H/R99G (H2Adm) and H2A.X-H38N/G99R (H2A.Xdm). This shows that the non-tail H2A.X-specific residues contribute to the increased thermal stability, whereby the H2Adm and H2A.Xdm nucleosomes dissociate at higher and lower temperatures, respectively, compared to the H2A and H2A.X counterparts (Fig. 4b). Nonetheless, the double mutants only partially recapitulate the thermal stability characteristics of the source variant, meaning that the H2Adm and H2A.Xdm nucleosomes are still less and more stable, respectively, than the H2A.X and H2A counterparts. This indicates that the other amino acid differences are also contributing to the elevated thermal stability of the (γ)H2A.X nucleosomes.

In order to assess nucleosome dynamics under near physiological conditions, we conducted micrococcal nuclease digestion experiments. Differences in the susceptibility of nucleosome towards digestion reflect distinctions in DNA accessibility that coincide with differences in the rates with which the double helix dissociates from and re-associates with the histone octamer. While H2A and H2A.X nucleosomes display very similar susceptibility towards digestion, that of the γH2A.X nucleosome is significantly higher, in particular for the earliest digest time points in which the nucleosome core DNA termini are initially degraded (Fig. 4c and Supplementary Fig. 6). This suggests that the intrinsic (37 °C) stability of the H2A and H2A.X nucleosomes is similar, while that of the γH2A.X nucleosome is reduced, with the DNA ends being more available to factor binding.

**Maximal catalytic efficiency with γH2A.X nucleosome.** Given that different core histones and tail elements have been shown to both inhibit and activate PARP1 catalysis in *Drosophila*[20,21], stimulation of PARylation activity by the different DSB constructs was measured. This reveals that PARP1 is activated by γH2A.X nucleosome to a substantially greater extent, in terms of overall catalytic efficiency ($k_{cat}/K_M$), than nucleosome composed of either H2A, H2A.X or the H2A.X-S139E mutant (Fig. 5a). The enhanced activation by γH2A.X nucleosome holds even relative to the corresponding 167 bp naked DNA, in spite of the availability of multiple PARP1 binding sites on the latter (Supplementary Fig. 7). The disproportionately high catalytic efficiency brought about through binding to γH2A.X nucleosome results from an especially low $K_M$ value (Fig. 5a).

In order to shed light on the dynamics of PARP1 interaction with DSB sites under conditions where PARylation is supported, we conducted BLI binding measurements in real time with buffers containing NAD$^+$ substrate. This reveals that DNA, even in the presence of NAD$^+$, cumulatively associates with PARP1 over a 300 s time course (1 nM DNA as analyte),

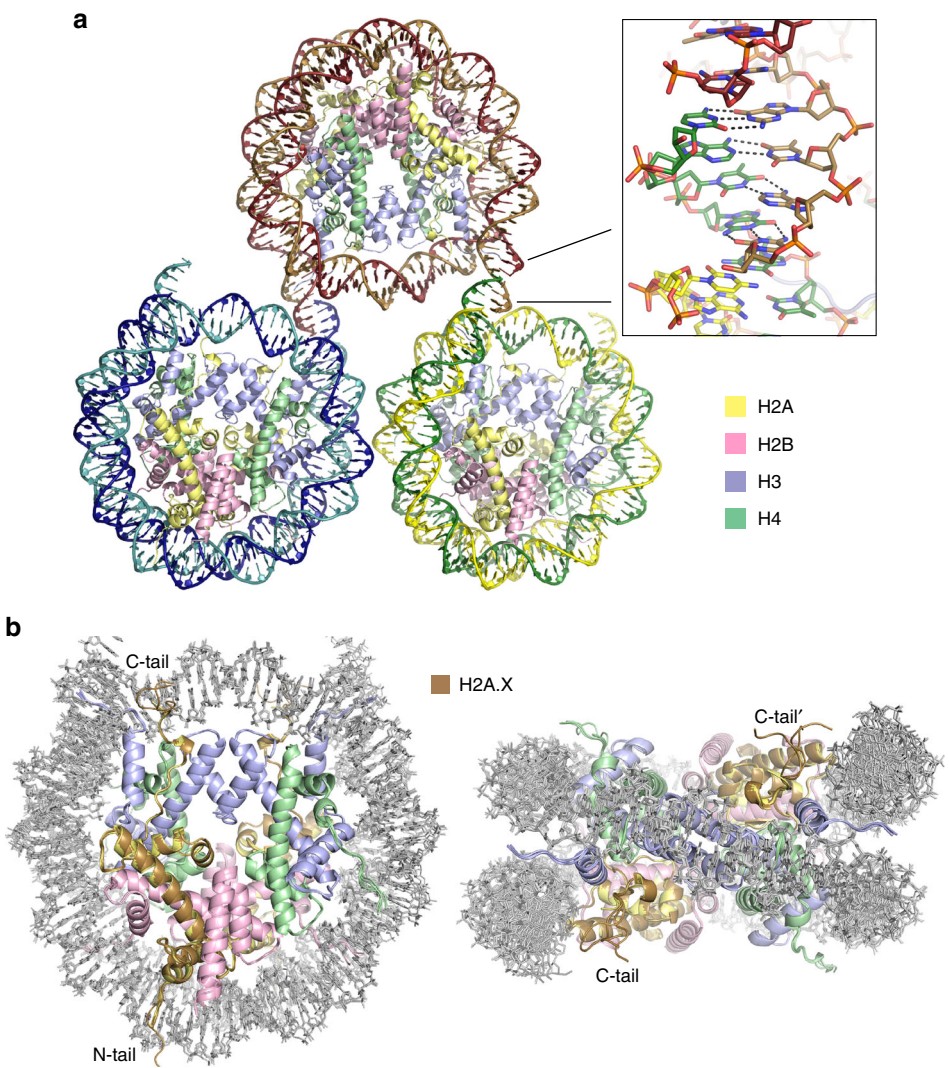

| | |
|---|---|
| H2A | yellow |
| H2B | pink |
| H3 | purple |
| H4 | green |

**Fig. 2 Structures of NCP composed of H2A and the H2A.X variant. a** Inter-nucleosomal annealing of the 4-nucleotide 3′ overhangs in the 147s-NCP structures (NCP assembled with H2A shown) yields continuous, 147 bp repeat, fibres composing the lattice. **b** Five NCP structures corresponding to superposition of the H2A.X-NCP, γH2A.X-NCP, H2A.X-S139E-NCP and H2A-147s-NCP models onto that of H2A-(145 bp)-NCP. **a**, **b** Histone proteins are coloured by type: H2A, yellow; H2A.X, bronze; H2B, pink; H3, purple; H4, green.

with dissociation apparent only upon switching to the analyte (DNA)-free condition (Fig. 5b). This is similar to the behaviour seen for nucleosome binding to PARP1 in the absence of $NAD^+$, with cumulative association over the entire 300 s time course (1 nM nucleosome as analyte). In contrast, for nucleosome in the presence of $NAD^+$, dissociation from PARP1 is seen well before completion of the 300 s time course. For H2A nucleosome, plateau and dissociation is apparent after about 170 s, whereas for either H2A.X and γH2A.X nucleosome, this occurs at about 125 s. For the H2A.X-S139E nucleosome, dissociation is apparent already at ~110 s.

Activation of PARP1 by H2A- or H2A.X-NUC167, in the presence of $NAD^+$ substrate, coincides exclusively with PARylation of PARP1, as opposed to the core histones (Supplementary Fig. 8). Therefore, auto-PARylation is seen to diminish PARP1 affinity for the nucleosomes, but not for naked DNA. Moreover, the speed of onset for auto-PARylation-induced dissociation from nucleosome is roughly correlated with the measured kinetic association rates (relative to H2A nucleosome, PARP1 associates 2.9, 4.5 and 9.1 times faster to γH2A.X, H2A.X and H2A.X-S139E nucleosomes, respectively; Fig. 1b and Supplementary Table 1).

## Discussion

The functional exploitation of nucleosome dynamics modulated via histone variant substitution and post-translational modifications is well documented[22,23]. Here, we have shown that, relative to canonical H2A nucleosome, PARP1 associates more rapidly and at higher affinity with nucleosomes composed of the H2A.X variant, with the maximal values observed coinciding with γH2A.X nucleosome (the S139E mutant notwithstanding). Considering that cells deposit H2A.X at locations that are most susceptible to endogenous DSB formation[11], this suggests that H2A.X nucleosomes may directly facilitate repair by fostering interaction with PARP1 (Fig. 6).

The thermodynamic characteristics of nucleosomes assembled with H2A and H2A.X variants are distinct, and the influence of the γH2A.X modification in particular stands out, which sheds light on its special function in DNA repair. We find that among these three histone species, H2A nucleosome displays higher salt stability than either H2A.X or γH2A.X nucleosome, but vice versa in terms of thermal stability. Indeed, others have shown that nucleosomes assembled with H2A.X or H2A.X treated with DNA-PK (protein kinase) have lower salt stability compared to H2A nucleosome[17] and that γH2A.X is more readily salt-

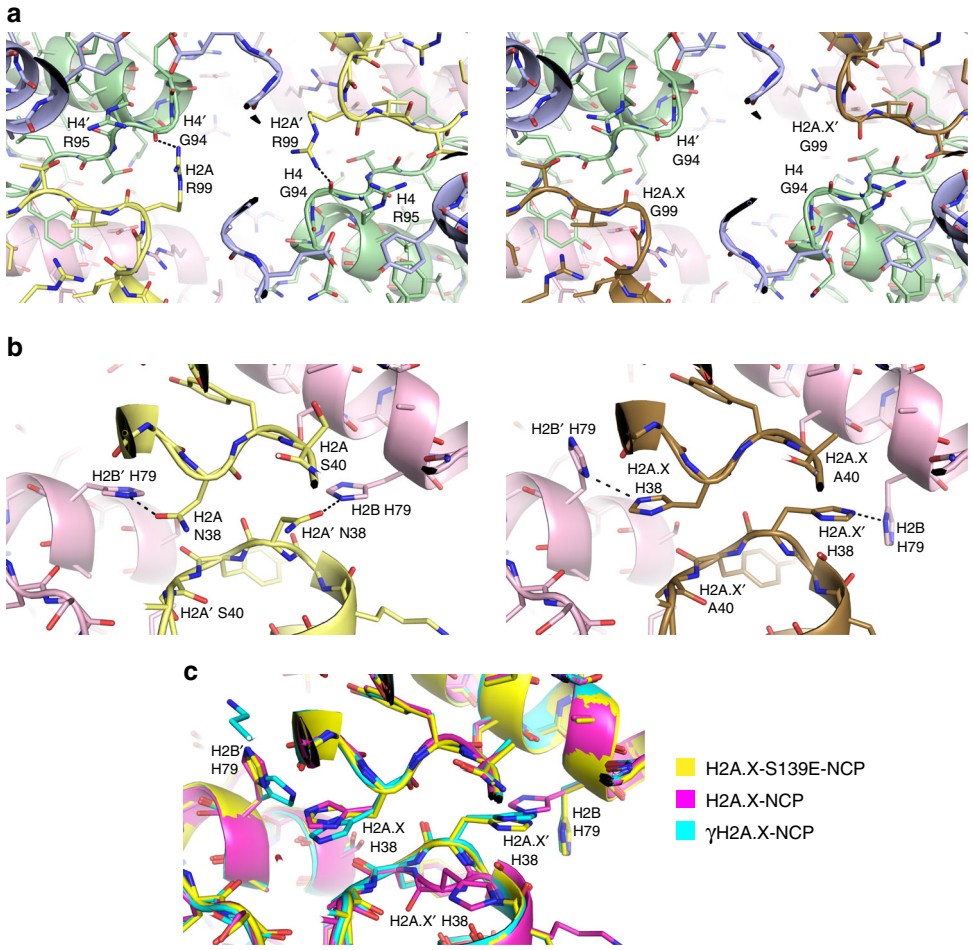

**Fig. 3 Structural differences between H2A and H2A.X nucleosomes. a** The dimer–tetramer–dimer interface around the centre of the histone octamer. **b** The dimer–dimer interface. **a**, **b** Carbon atoms of histone proteins are coloured by type: H2A (left side), yellow; H2A.X (right side), bronze; H2B, pink; H3, purple; H4, green. **c** Comparison of dimer–dimer interface conformation between the three H2A.X-NCP structures.

dissociated from chromatin in situ[18]. On the other hand, the latter study also observed no significant difference in H2A versus H2A.X release from chromatin induced by either high salt or treatment with nickase/DNase I[18]. While we find that the two non-tail residue changes that distinguish H2A from H2A.X (amino acids 38 and 99) contribute significantly to the relative differences in thermal stability, other residues that differ between the variants are also involved in the effect. One of the two non-tail residue distinctions, H38 in H2A.X, results in an unusual histidine-pairing motif at the L1–L1′ interface of the H2A.X-H2B dimers. Since these two residues can interact only repulsively when both protonated, they present the interesting possibility of a variant-specific pH-sensitive mechanism that could impact nucleosome dynamics—for instance, playing out within the acidic milieu of cancer cells.

The nucleosome unfolding measurements by high salt and high temperature coincide with distinct mechanisms of inducing dissociation, with the former diminishing electrostatic forces while enhancing the hydrophobic effect and the latter augmenting entropic effects. Thus, the reverse trend observed between the two methods suggests that nucleosomes assembled with the two variant types have stability features that differ with respect to the relative contributions of electrostatic and non-electrostatic interactions. Indeed, most of the amino acid changes between H2A and H2A.X involve substitutions of charged (if one includes histidine) for uncharged side chains, in addition to the longer C-terminal tail of H2A.X, which entails three (four for γH2A.X)

more charged groups. Nonetheless, the micrococcal nuclease digestion assay shows that double helix accessibility is very similar between the H2A and H2A.X nucleosomes, whereas that of the γH2A.X nucleosome is much increased. In fact, an earlier study showed that nucleosome assembled with DNA-PK-treated H2A.X displays (relative to untreated H2A.X nucleosome) enhanced H2A.X-H2B dimer exchange catalysed by FACT as well as elevated sensitivity towards restriction enzymes and DNA methyltransferase[19]. Therefore, we conclude that γH2A.X nucleosomes have explicit dynamic properties that render the double helix with greater access towards factor binding. This would account for the distinctly fast association and high affinity of PARP1 in binding to γH2A.X nucleosome.

Given the recent finding that PARP1 binding induces partial unfolding of the nucleosome core[24], it is likely that the dynamic alterations conferred by the γH2A.X modification facilitate association and stabilize the bound state by rendering the nucleosome more susceptible to conformational remodelling. Increased dynamics would account for more rapid association rates and higher affinity by reducing the energetic penalty associated with structural modification in the PARP1-bound state.

PARP1 displays faster binding to and higher affinity for all of the H2A.X-type nucleosomes compared to the H2A nucleosome. But given that the nuclease sensitivity (DNA accessibility) between the H2A and H2A.X nucleosomes is about the same and also that the unique catalytic effect from activation by γH2A.X nucleosome is not recapitulated with the H2A.X-S139E

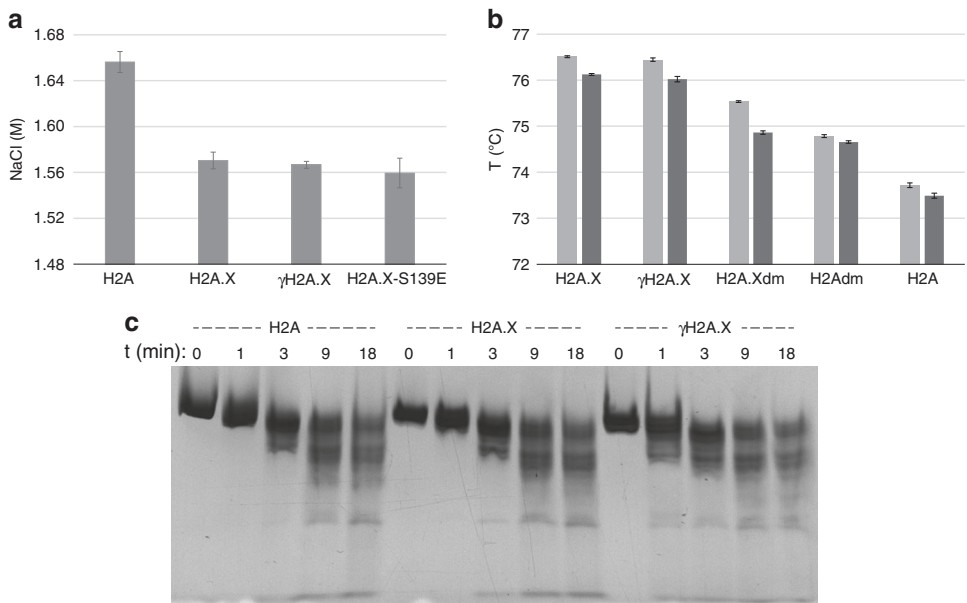

**Fig. 4 H2A and H2A.X nucleosomes display distinct stability/dynamics characteristics. a** Salt-induced histone–DNA dissociation measured by tyrosine fluorescence spectroscopy (mean ± s.d., $n = 2, 3, 3, 2$ independent experiments). **b** High temperature-induced histone–DNA dissociation based on either hydrophobic dye binding fluorescence (light grey) or DNA denaturation (dark grey; mean ± s.d., $n = 2$ independent experiments). **c** Micrococcal nuclease digestion analysis of H2A-NCP, H2A.X-NCP and γH2A.X-NCP. After incubation with nuclease for the specified times, samples were visualized by native DNA PAGE subsequent to degradation of proteins. A digest profile together with a 10 bp DNA ladder is shown in Supplementary Fig. 6. Source data are provided as a Source Data file.

phosphorylation mimic, one should not rule out that specific or non-specific interactions between PARP1 and histone elements contribute to the differential binding and catalytic phenomena. However, in any case we observe an absence of substantial interaction between PARP1 and histone dimers, tetramers or (dissociated) octamer in isolation, and this is consistent with analogous measurements based on the HI-FI FRET method, which showed strong interactions towards H2A-H2B dimer and H3-H4 tetramer only with auto-modified (PARylated) PARP1[15]. Indeed this is likely involved in the mechanism for the auto-PARylation-mediated dissociation of PARP1 from the nucleosomes that we observe in the dynamic binding assays (Fig. 5b), whereby PARylated-PARP1-histone interactions facilitate dissociation from the DSB.

PARP1 accumulation at DNA lesion sites has been shown to occur immediately upon induction of DNA damage (micro-irradiation), with a time required for half-accumulation measured at only 1.6 s (ref. [13]). In this regard, association kinetics accelerated through H2A.X variant deposition would help to promote this first key step in the repair process. Correspondingly, evidence has suggested that PARP1 auto-modification triggers its dissociation from the chromatin template in mammalian cells[13] as well as for other systems[20,25]. Moreover, auto-modified PARP1 has been observed to have reduced affinity for nucleosomal, but not naked DNA, constructs in vitro[15]. Additionally, the partial nucleosome core unfolding characterized by PARP1 binding is fully reversed by auto-modification-induced dissociation of the enzyme[24]. Consistent with these findings, we observe that PARP1 activation by nucleosomal, but not naked DNA, templates promotes dissociation of the enzyme, and that the onset of dissociation coincides with the rate of PARP1 binding. This suggests that H2A.X deposition into chromatin, on top of histone-packaging in general, can facilitate both the association and the ultimate release of PARP1 in the DNA repair process.

A recent study demonstrated the importance of PARylation activity per se, wherein a PARP1 mutant with compromised catalytic function displayed a severely hampered DNA damage response in mouse[26]. This emphasizes the requirement for timely catalytic output by PARP1 and could help explain why its activity could be propitiously modulated by the nature of the DNA damage template. Although $NAD^+$ levels in the cell are typically substantial, they have been shown to drop precipitously following PARP1 activation[27]. Within 15 min of activation, $NAD^+$ concentration was found to decrease by 80%, and after 30 min, $NAD^+$ was no longer detectable. Furthermore, ATM-mediated phosphorylation of H2A.X, yielding γH2A.X, occurs subsequent to PARP1 localization at DNA damage sites, and this localization can persist for at least 30 min[13]. In this sense, our finding that PARP1 activation by γH2A.X nucleosome elicits especially high catalytic efficiency, stemming from a particularly low $K_M$ value, suggests that the presence of γH2A.X at damage sites may help provide a final kick towards PARylation when the $NAD^+$ concentration drops to low levels (Fig. 6).

## Methods

**PARP1 and NPARP1**. *Homo sapiens* PARP1 and NPARP1 (amino acids 1–486) coding DNA was obtained from the Protein Production Platform (NTU, Singapore) and cloned into pET28a bacterial expression vector. The 6×His-tagged proteins were overexpressed in *Escherichia coli* RIPL cells overnight at 18 °C, and purification was carried out at 4 °C. Cells were harvested and resuspended in buffer A (20 mM Tris-HCl [pH 7.5], 5% glycerol, 500 mM NaCl, 1 mM β-mercaptoethanol, 0.5 mM phenylmethylsulfonyl fluoride and 0.05% (v/v) protease inhibitor cocktail; Calbiochem, San Diego, CA, USA) and sonicated, followed by centrifugation at 50,000 × g for 20 min. The supernatant was loaded onto a 5 ml IMAC-Ni column (GE Healthcare, Chicago, IL, USA) pre-equilibrated with buffer A. The protein was eluted with a linear gradient of 0–500 mM imidazole in buffer A. Fractions containing the protein of interest were pooled together and diluted with buffer A, lacking the NaCl, in order to reduce the NaCl concentration to 250 mM. The sample was loaded onto a 5 ml heparin column pre-equilibrated with buffer A containing 300 mM NaCl. Protein was eluted with a gradient of 0.3–1.0 M NaCl in buffer A. Fractions containing the protein were pooled together and concentrated to 2 ml, followed by separation on a Superdex-200 column (GE Healthcare, Chicago, IL, USA) in buffer B (20 mM Tris-HCl [pH 7.5], 5% glycerol, 200 mM NaCl and 1 mM β-mercaptoethanol). Peak fractions containing >99% pure protein, with $A_{260}/A_{280}$ <0.6, were pooled together and stored at −80 °C.

## a

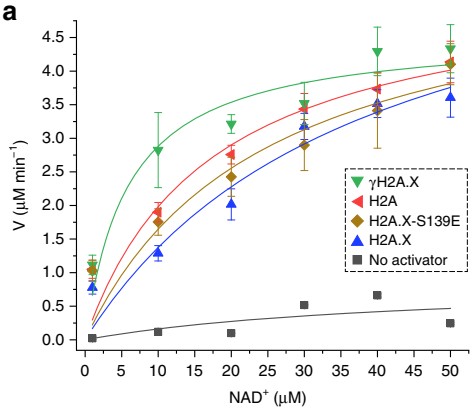

| Activator | $V_{max}$ ($\mu$M min$^{-1}$) | $K_M$ ($\mu$M) | $k_{cat}$ (min$^{-1}$) | $k_{cat}/K_M$ (min$^{-1}\mu$M$^{-1}$) |
|---|---|---|---|---|
| $\gamma$H2A.X | 4.6 ± 0.4 | 5.9 ± 2.7 | 664.5 | 112.9 |
| H2A | 5.4 ± 1.0 | 17.2 ± 8.5 | 782.8 | 45.6 |
| H2A.X-S139E | 5.7 ± 1.6 | 24.5 ± 15.7 | 825.0 | 33.7 |
| H2A.X | 6.6 ± 2.1 | 38.2 ± 23.3 | 962.7 | 25.2 |
| No activator | 0.9 ± 1.4 | 44.5 ± 126.7 | 128.1 | 2.9 |

## b

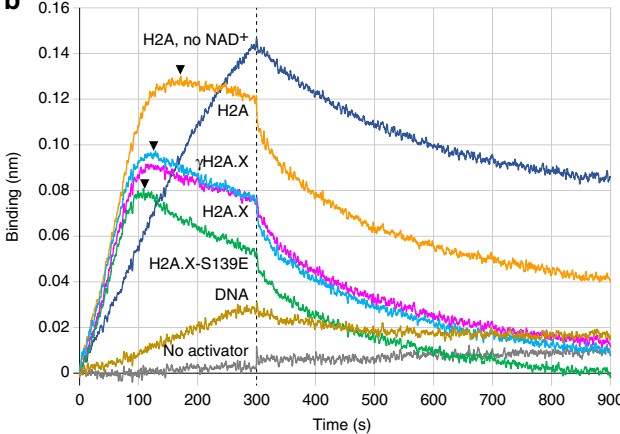

**Fig. 5 Influence of activator binding on PARP1 catalytic activity and vice versa. a** Catalytic profile and parameters for PARP1 activation by H2A and H2A.X nucleosomes (mean ± s.d., $n = 2$ independent experiments). **b** Effect of auto-PARylation on activator binding behaviour. BLI measurements were performed in the presence of 100 $\mu$M NAD$^+$ (with exception of uppermost sample), with PARP1 and either no activator, naked DNA or H2A or H2A.X nucleosome. Probes were switched to analyte(activator)-free conditions after 300 s (dashed line). Arrowheads indicate approximate inflexion points, coinciding with observable dissociation, for H2A and H2A.X nucleosome in the presence of NAD$^+$. Source data are provided as a Source Data file.

**$\gamma$H2A.X synthesis.** $\gamma$H2A.X [H2A.X (pS139)] was synthesized using a ligation-desulfurization strategy (Supplementary Fig. 9a)[28]. Briefly, the C$^\alpha$-carboxy thioester of H2A.X (amino acids 1–134) was first prepared by thiolysis of the H2A.X (1–134)-intein-chitin-binding domain (CBD) fusion protein with MESNa (sodium 2-mercaptoethanesulfonate) via intein-mediated protein splicing[29]. The MES (2-mercaptoethanesulfonate) thioester of H2A.X (1–134) was then ligated with the synthetic C-terminal octapeptide containing the phosphorylated-Ser residue, H-CTQApSQEY-OH, to form the full-length $\gamma$H2A.X containing an Ala-Cys mutation at position 135, H2A.X (A135C, pS139). To be noted is the formation of a minor side product (observed MW, 14,119 Da) due to the spontaneous lactamization of the C-terminal Lys residue of the H2A.X (1–134) thioester (Supplementary Fig. 9b). This side reaction likely started as soon as the H2A.X (1–134) thioester was formed, as the lactam side product was already present in the thioester material before the ligation reaction. It was not possible to remove this side product by reverse-phase high-performance liquid chromatography (HPLC) as it co-eluted with the desired ligation product, H2A.X-A135C-pS139. However, since the ligation product contains a free Cys residue, oxidative dimerization led to the formation of an H2A.X-A135C-pS139 dimer, the much larger size of which

made it easier to remove the lactam side product. Desulfurization was then conducted on the purified H2A.X-A135C-pS139 at 37 °C for 32 h in the presence of tris(2-carboxyethyl)phosphine (TCEP) and glutathione using 2,2′-azobis[2-(2-imidazolin-2-yl)propane]dihydrochloride (VA-044) as the radical initiator. The final product was purified by reverse-phase HPLC using a C4 semi-preparative column.

Amino acid derivatives, coupling reagents and resins were purchased from Chemimpex, Novabiochem and GL Biochem (Shanghai, China). All other chemical reagents were purchased from the Sigma-Aldrich Chemical Company, Alfa Aesar and Acros Organics. Analytical HPLC analyses were performed using a Shimadzu system equipped with a photodiode array detector and an analytical C18 or C4 reverse-phase column (5 U, 250 × 4.6 mm$^2$) at a flow rate of 1.0 ml/min. Purification was performed using a semi-preparative HPLC column on a Shimadzu system. The HPLC buffers utilized consisted of buffer C (H$_2$O containing 0.045% trifluoroacetic acid [TFA]) and buffer D (90% acetonitrile in H$_2$O, containing 0.039% TFA). The temperature of the HPLC column chamber was maintained at 40 °C. Peptide/protein masses were measured on a Thermo FINNIGAN LTQ Deca XP MAX equipped with an ESI ion source or on a 4800 MALDI-TOF/TOF Analyser operating in mass spectrometer (MS) reflector-positive ion mode and using α-cyano-4-hydroxycinnamic acid as the matrix.

Fmoc-Tyr(tBu) was loaded to 2-chloro-trityl resin for the preparation of H-Tyr (tBu)-O-trityl resin. To a mixture of 2-chlorotrityl chloride resin (0.2 g, 1.1 mmol/g) and Fmoc-Tyr(tBu)-OH (0.4 g) in 5 ml of dry dichloromethane (DCM), 0.23 ml of diisopropylethylamine (DIEA) was added. The mixture was shaken at room temperature (rt) overnight. Then the DCM was removed. The resin was then washed with DCM, N,N′-dimethylformamide (DMF) and treated with 20% piperidine in DMF (1 × 2 min, 1 × 18 min) at rt, followed by washing with DCM and DMF to yield the H-Tyr(tBu)-O-trityl resin.

A reaction vessel containing H-Tyr(tBu)-O-resin was loaded on a Liberty I CEM microwave peptide synthesizer. The method used for Fmoc removal was a treatment of 20% piperidine with 0.1 M 1-hydroxybenzotriazole in DMF at 60 °C (1 × 30 s, 1 × 180 s). The amino acid coupling method used was a treatment with 2.5 ml of 0.2 M Fmoc-(amino acid) in DMF, 1 ml of 0.5 M benzotriazol-1-yl-oxytripyrrolidino-phosphonium hexafluorophosphate in DMF and 0.5 ml of 2 M DIEA in DMF at 60 °C for 10 min. All amino acids were coupled twice. The Fmoc amino acids with side chain-protecting groups were Fmoc-Glu(OtBu)-OH, Fmoc-Gln(Trt)-OH, Fmoc-Ser(PO(OBzl)OH)-OH, Fmoc-Thr(tBu)-OH and Fmoc-Cys (Trt)-OH. The fully protected peptide resin was obtained at the end.

After Fmoc removal using 20% piperidine in DMF (1 × 2 min, 1 × 18 min), the peptide was cleaved from the resin by a cocktail containing 2.5% (v/v) H$_2$O, 2.5% (v/v) triisopropylsilane, 5% (v/v) ethanedithiol and 90% (v/v) TFA at rt for 1.5 h. Diethyl ether was added to the cleavage mixture to achieve peptide precipitation. The crude peptide was then purified by reverse-phase HPLC (C18 prep, 0–40% buffer D in buffer C) and characterized by MS (electrospray ionization). H-CTQApSQEY-OH, found (m/z): 1010.96 [M + H]$^+$, 506.34 [M + 2 H]$^{++}$, calcd: 1009.95 [M + H]$^+$.

*Homo sapiens* H2A.X expression plasmid (Genescript, NJ, USA) was used as a template to PCR amplify the H2A.X (1–134) coding sequence using the following forward and reverse primers, respectively:

GGAATTCATATGAGCGGCCGTGGTAAAACCGGTGG
CCTTAACTAGTGCATCTCCCGTGATGCATTTTTTACCGCCGCTTGGGGC.

The H2A.X (1–134) coding fragment was cloned into pTXB1 plasmid (NEB, MA, USA) in frame with GyrA intein and the CBD at the C-terminus (Supplementary Fig. 9a). *Escherichia coli* RIPL cells were used to express the H2A.X (1–134)-intein-CBD fusion protein using auto-induction media. Harvested cells were resuspended in buffer E (20 mM Tris [pH 8.5], 2 M urea, 500 mM NaCl and 0.2% Triton-X) and lysed by sonication, followed by centrifugation at 20,000 × g for 15 min at 4 °C. The supernatant was loaded onto Chitin Beads (Bio-Rad, CA, USA) pre-equilibrated with buffer E. The chitin beads were washed with 10 column volumes of buffer E. The beads were incubated with buffer E containing 50 mM MESNa overnight at 4 °C to induce intein cleavage. The cleaved H2A.X (1–134), with a reactive thioester at the C-terminus, was eluted in buffer E. The sample was further purified with a heparin column pre-equilibrated with buffer F (20 mM Tris [pH 7.5], 2 M urea, 5% glycerol). The protein was eluted using a gradient of 0.5–1 M NaCl in buffer F. Peak fractions containing H2A.X (1–134) thioester were pooled together. Matrix-assisted laser desorption/ionization-time of flight (MALDI-TOF) was used to confirm the molecular weight of the purified protein.

The MES thioester of H2A.X (1–134), that is, H2A.X (1–134)-CO-SCH$_2$CH$_2$SO$_3^-$ (24 mg), and the C-terminal octapeptide, H-CTQApSQEY-OH (3 mg), were dissolved in 323 $\mu$l of 6 M guanidine-HCl solution (0.2 M phosphate buffer, pH 8). To this mixture was added 13.9 $\mu$l of 0.727 M TCEP in H$_2$O (pH adjusted to ~7 using NaOH solution) and 10.1 mg of sodium thiophenolate. The final concentrations of each component were 5 mM for H2A.X (1–134) thioester, 8.83 mM for the C-terminal peptide, 30 mM for TCEP and 3% (w/v) for sodium thiophenolate. The resulting mixture was kept shaking at rt for 24 h, and the product was then purified by reverse-phase HPLC (C4 semi-prep, 5–70% buffer D in buffer C). The desired ligation product, H2A.X (A135C, pSer139), was characterized by MS (ESI), found: 15127 [M + H]$^+$, calcd: 15126.3 [M + H]$^+$ (Supplementary Figs. 10–12).

A minor impurity (with a found m/z 14119 [M + H]$^+$; see above for description; Supplementary Figs. 10–12) could not be removed by HPLC

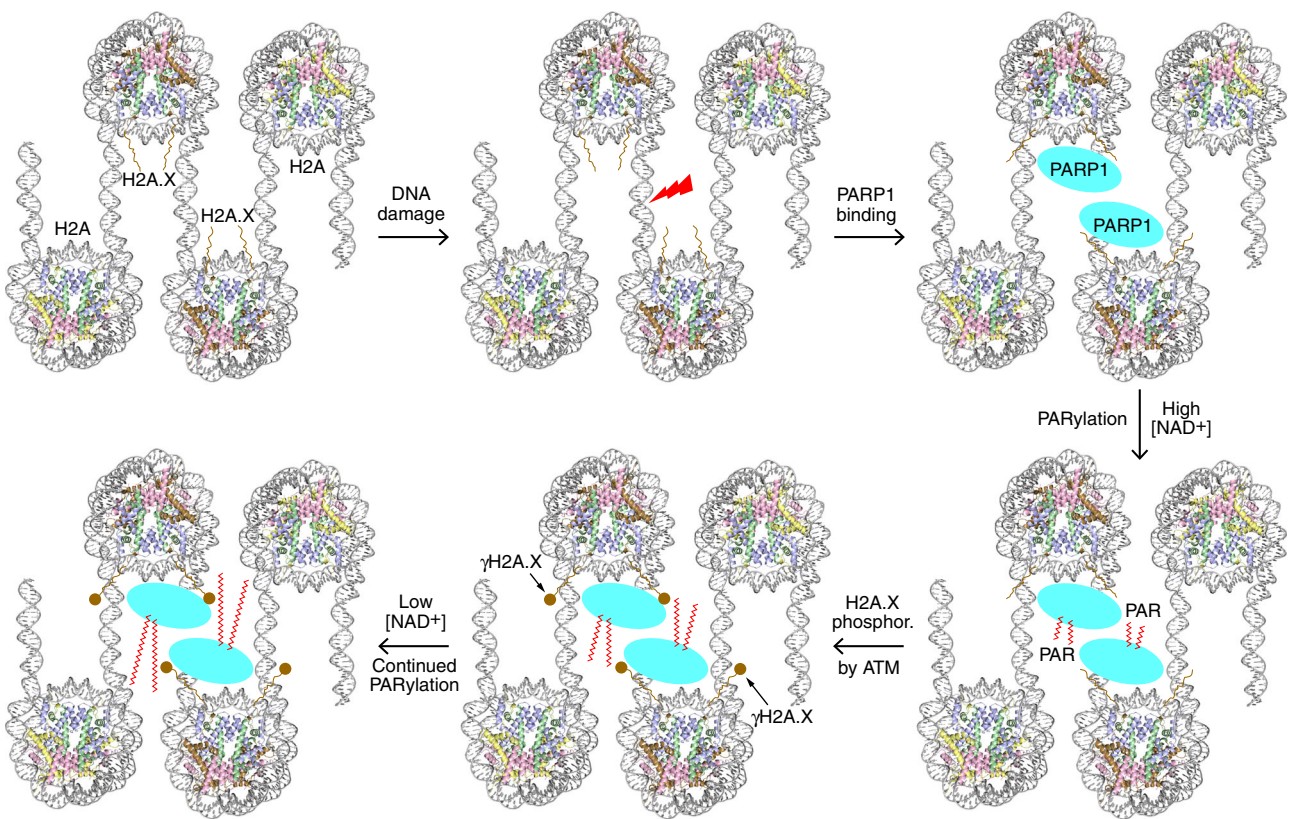

**Fig. 6 Model for the contribution of H2A.X nucleosomes towards PARP1-mediated DNA repair.** A DNA damage hotspot is demarcated by H2A.X deposition. Subsequent to a double helix breakage event, PARP1 rapidly associates with the H2A.X-nucleosomal DSBs at high affinity. This ensures prompt activation of PARP1, initiating PARylation, and H2A.X is subsequently phosphorylated to γH2A.X, which further stabilizes the assembly. Even as the NAD⁺ concentration drops to low levels, the influence of γH2A.X supports continued PARylation.

purification. To remove the lactam impurity, sample was dissolved in 50 mM Tris (pH 7.5), 150 mM NaCl under non-reducing condition. The desired product, H2A. X (A135C, pSer139), was found to form disulfide-bonded dimer, whereas the unligated (lactam) side product could not dimerize due to the lack of a thiol group. Accordingly, the dimer was purified from the lactam side product using size-exclusion chromatography (Superdex 75 column; GE Healthcare, Chicago, IL, USA).

TCEP (66.32 mg) was dissolved in 0.5 ml of 0.2 M phosphate buffer (pH 6) containing 6 M guanidine-HCl. To this solution, 57 μl of 8 M NaOH was added to adjust the pH to around 5–6. This TCEP solution was then used to dissolve ~5 mg H2A.X (A135C, pSer139). To the resulting solution, 9.92 μl of 0.1 M of glutathione (reduced form) in $H_2O$ was added, followed by 66.11 μl of 0.2 M VA-044 in $H_2O$. The mixture was incubated at 37 °C for 12 h, and the reaction checked with ESI-MS, which showed the desired product H2A.X (pS139) and the starting material H2A.X (A135C, pSer139). A freshly prepared glutathione solution (9.92 μl, 0.1 M) was added, followed by a freshly made VA-044 solution (66.11 μl, 0.2 M), and the pH of the reaction mixture was readjusted to around 6–6.5. The resulting mixture was continuously incubated at 37 °C for an additional 20 h, and the reaction was checked again by ESI-MS, which showed much improvement with almost no starting material left. The reaction mixture was then subjected to purification by reverse-phase HPLC (C4 semi-prep, 5–70% buffer D in buffer C) to give the desired product, γH2A.X [H2A.X (pS139)], as confirmed by ESI-MS [found ($m/z$): 15096.2 $[M + H]^+$, calcd: 15094.3 $[M + H]^+$] (Supplementary Figs. 13–15).

**Nucleosome assembly**. Nucleosome and NCP were assembled with recombinant *H. sapiens* core histones and various recombinantly produced DNA fragments[30–32], including the previously described palindromic 145 bp 601L[33] and α-satellite[34] derivatives and newly designed sequences (described below). *Homo sapiens* H2A.X (gene synthesis by Genescript, NJ, USA) was cloned into pHCE vector, and PCR-based site-directed mutagenesis was used to generate the H2A.X-S139E mutant. The H2Adm (N38H, R99G) and H2A.Xdm (H38N, G99R) expression constructs were cloned into pET28a vector (Genescript, NJ, USA). The core histones, with N-terminal hexahistidine tags, were expressed in BL21(DE3) (H2A, H2Adm, H2A. X, H2A.Xdm, H2B and H3) or JM109(DE3) (H4) *E. coli* cells and affinity purified with a HiTrap IMAC FF column (GE Healthcare, Chicago, IL, USA). The N-terminal His-tag was removed by thrombin digestion, and histones were purified further by fast protein liquid chromatography using Resource S (H2A, H2Adm,

H2A.X, H2A.Xdm, H2B and H3) or Mono S (H4) columns (GE Healthcare, Chicago, IL, USA).

The sticky-end 147s DNA fragment (used to solve the NCP-γH2A.X) was designed based on calculated length-twist relationship that would allow annealing of the compatible 4-nucleotide overhangs at each DNA terminus with that of another NCP throughout the lattice. The 147s consists of a 143 bp core sequence, which is a 'de-palindromized' version of the palindromic 145 bp left-half Widom 601 sequence[35] (i.e. 601L[33]). The 147s repeats are connected by *Kpn*I restriction sites in the expression clone (pUC19 vector; Genescript, NJ, USA), yielding the palindromic 4-nucleotide 3′ overhangs upon excision from the plasmid.

147s-Strand A
CATATATCCCGGTGCCGAGGCCGCTCAATTGGTCGTAGACAGCTCTA
GCACCGCTTAAACGCACGTACGCGCTGTCTACCGCGTTTTAACCGCCACT
AGAAGCGCTTACTAGTCTCCAGGCACGTGTGAGACCGGCATATATGGTAC
147s-Strand B
CATATATGCCGGTCTCACACGTGCCTGGAGACTAGTAAGCGCTTCT
AGTGGCGGTTAAACGCGGTAGACAGCGCGTACGTGCGTTTAAGCGGT
GCTAGAGCTGTCTACGACCAATTGAGCGGCCTCGGCACCGGGATATAT
GGTAC

Likewise, the (blunt-end) 155 and 167 bp DNA fragments consisted of the same 145 bp core sequence, with the addition of a 10 bp linker section on one terminus or an 11 bp linker segment at either terminus (repeats cloned into pUC57 vector; EZbiolabs, Carmel, IN, USA).

155-Strand A
ATCACAATCCCGGTGCCGAGGCCGCTCAATTGGTCGTAGACAGCTCT
AGCACCGCTTAAACGCACGTAGGGAATGTTTGTTCTTATTTAAGCGCAC
CTAGAGCTCGCTACTCGCATTCTACGATCCGCAAGGGATATTTGGAGAA
AAAAAACGAT
155-Strand B
ATCGTTTTTTTTCTCCAAATATCCCTTGCGGATCGTAGAATGCGAGTA
GCGAGCTCTAGGTGCGCTTAAATAAGAACAAACATTCCCTACGTGCGT
TTAAGCGGTGCTAGAGCTGTCTACGACCAATTGAGCGGCCTCGGCACC
GGGATTGTGAT
167-Strand A
ATCTACCGGTTCGCACAATCCCGGTGCCGAGGCCGCTCAATTGGTC
GTAGACAGCTCTAGCACCGCTTAAACGCACGTACGGAATCCGTACGTGC
GTTTAAGCGGTGCTAGAGCTGTCTACGACCAATTGAGCGGCCTCGGCAC
CGGGATTGTGCGAACCGGTAGAT

167-Strand B
ATCTACCGGTTCGCACAATCCCGGTGCCGAGGCCGCTCAATTGGTCG
TAGACAGCTCTAGCACCGCTTAAACGCACGTACGGATTCCGTACGTGCG
TTTAAGCGGTGCTAGAGCTGTCTACGACCAATTGAGCGGCCTCGGCACC
GGGATTGTGCGAACCGGTAGAT

For producing biotinylated nucleosome, purified DNA fragments were biotinylated using either a chemical or enzymatic method. For chemical derivatization, the DNA was treated with EDC (1-ethyl-3-[3-dimethylaminopropyl] carbodiimide) and imidazole, followed by incubation with hydrazide-PEG14-biotin, as specified by the vendor (Thermo Fisher Scientific, MA, USA). For enzymatic labelling of DNA, biotin-14-dCTP was coupled to the 3′ DNA end using terminal deoxyribonucleotide transferase. Biotinylated DNA was ethanol-precipitated and utilized for nucleosome reconstitution.

**SPR-based measurements**. All SPR-based experiments were performed on a Biacore 3000 instrument (GE Healthcare) at rt using a flow rate of 10 μl/min. This includes all of the measurements with NPARP1, as well as those of PARP1 interaction with DNA, NCP, NUC155 and NUC167. Biotinylated DNA or nucleosome/NCP (600–700 RU) having biotinylated DNA were immobilized on CM5 chips with pre-immobilized neutravidin. Twenty mM Tris (pH 7.5) with 150 mM NaCl was used as the running buffer. The reference cell was left unmodified to serve as a blank. Kinetic titrations were used to examine the nucleosome–PARP1 interactions. Different concentrations of PARP1 (2-fold serial dilution, 1.15–18.5 nM) or NPARP1 (2-fold serial dilution, 2.32–37 nM) were consecutively injected over the surface for 2.5–5 min and allowed to dissociate for 5–30 min. The resulting sensorgrams were corrected against the reference cell as well as buffer blanks and analysed with BiaEvaluation 3.1 (GE Healthcare) using kinetic titration 1:1 fit as a model.

**BLI-based measurements**. BLI-based measurements were carried out on an Octet RED96e instrument (Pall Fortebio, CA, USA) system at 25 °C, which was used to measure the parameters for interaction of PARP1 with NUC167 composed of either H2A, H2A.X, H2A.X-139E or γH2A.X. Biotinylated nucleosome was captured on Streptavidin Dip and Read[TM] Biosensors (Pall Fortebio, CA, US) with a threshold of 0.15–0.25 nm. For association kinetics measurement, the probes were dipped into wells containing varying concentrations of PARP1 (2-fold serial dilutions, 0.195–3.125 nM) in PBST buffer (8 mM Na_2HPO_4, 150 mM NaCl, 2 mM KH_2PO_4, 3 mM KCl, 0.05% Tween-20 [pH 7.4]) for 120 s, followed by dipping into wells containing PBST buffer for 15 min to measure the dissociation kinetics. For the PARP1–histone interactions measurements, His-tagged PARP1 was captured on anti-His Dip and Read[TM] Biosensors (Pall Fortebio, CA, USA) and dipped into wells containing 0.1–1 μM histone or histone assembly, and association and dissociation profiles were recorded in the same fashion. For reference, one of the probes was dipped into PBST during the association and dissociation phases. The data were analysed and fit to a 1:1 model using ForteBio Data Analysis 10 (Pall Fortebio, CA, USA).

For measuring the dynamics of PARP1 interaction with DNA/nucleosome under conditions where PARylation is supported, His-tagged PARP1 was captured on anti-His biosensors and dipped into wells containing 1 nM DNA/nucleosome, in the presence or absence of 100 μM NAD+. The association phase (permitting PARylation-induced dissociation) was recorded for 5 min, followed by dipping the biosensors in PBST buffer-containing wells to monitor dissociation for 10 min.

**Salt-induced nucleosome dissociation assay**. High salt-induced nucleosome dissociation was monitored by the increase in fluorescence resulting from the loss in histone protein tyrosine fluorescence quenching through interaction with the DNA[33,36,37]. Nucleosome (assembled with the 167 bp DNA fragment) samples (0.5 μM) were incubated with varying amounts of salt (0–2.4 M NaCl) in buffer containing 20 mM Tris (pH 7.5), 1 mM EDTA and 1 mM dithiothreitol (DTT) at 21 °C. Fluorescence readings were recorded with a Varian Cary Eclipse spectro-photometer (Agilent Technologies, Santa Clara, USA) at 305 nm (emission wavelength) upon excitation at 275 nm. Readings were normalized to a scale of 0 and 1 for values at 0 and 2.4 M NaCl, respectively. Data were analysed and fit to a sigmoidal curve using OriginPro 2018b (OriginLab Corp, Northampton, MA, USA). The salt concentration corresponding to 50% nucleosome dissociation was derived from the sigmoidal fit.

**Temperature-induced nucleosome dissociation assays**. Thermally induced nucleosome dissociation was measured by two different methods. Histone disassembly from the nucleosome was monitored via fluorescence from protein binding of the hydrophobic dye, SYPRO Orange[38]. Nucleosome (NUC167, 2.5 μM) was incubated with 10X SYPRO Orange dye (Thermo Fisher Scientific, MA, US) in a buffer containing 20 mM Tris (pH 7.5) and 200 mM KCl. Fluorescence was measured at 0.5 °C intervals over a temperature range of 25–95 °C (1 °C/min gradient) using a CFX96 Touch Real-Time PCR detection system (Bio-Rad, CA, USA). Data were normalized and analysed using OriginPro 2018b software (OriginLab Corp, Northampton, MA, USA). The temperature at which 50% of the total fluorescence increase is achieved ($T_m$) was calculated using a sigmoidal fit.

The second method employed is based on measuring hyperchromicity accompanying DNA denaturation as the nucleosome dissociates[39–41]. Absorbance at 257 nm was monitored using a Varian Cary 300 Bio UV/Vis spectrophotometer (Agilent Technologies, Santa Clara, USA). Nucleosome (NUC167, 0.5 μM) in 20 mM Tris (pH 7.5) and 200 mM KCl was subjected to increasing temperature over the range of 25–95 °C (1 °C/min gradient) while monitoring $A_{257}$ at 0.5 °C intervals. Absorbance readings were normalized and processed using the OriginPro 2018b software (OriginLab Corp, Northampton, MA, USA). Sigmoidal fitting was used to determine the $T_m$ values.

**Micrococcal nuclease digestion assay**. NCP (0.5 μM) was incubated at 37 °C with 0.5 μl micrococcal nuclease (2000 gel unit/μl; NEB, MA, USA) in CutSmart buffer (50 mM potassium acetate, 20 mM Tris-acetate [pH 7.9], 10 mM magnesium acetate, 100 μl/ml bovine serum albumin) supplemented with 2.5 mM CaCl_2. At the time points indicated in the figures, digestion was arrested by the addition of EDTA to a final concentration of 50 mM, followed by degradation of proteins via incubation with 0.5 U of proteinase K (Thermo Fisher Scientific, MA, USA) for 10 min. The samples were analysed using 6% native PAGE. Four independent experiments comparing the micrococcal nuclease sensitivity of H2A, H2A.X and γH2A.X nucleosomes were carried out, which all show the same qualitative characteristics.

**PARP activity assays**. Varying concentrations of Biotinylated NAD+ (BP Biosciences, San Diego, CA, USA) were incubated with 1 ng/μl PARP1 (~8 nM) in assay wells containing immobilized H2A-H2B dimer, with the presence or absence of 10 nM nucleosome/DNA as activator. The assay mixture was removed at different time points (1, 5, 10 and 20 m), and the wells were washed and developed using a 1:4000 dilution of horseradish peroxidase (HRP)-conjugated streptavidin (Thermo Fisher Scientific, MA, USA), followed by further washing and incubation with HRP substrate (Promega, WI, USA). The absorbance was measured at 450 nm, subsequent to H_2SO_4 treatment, using a Tecan Infinite M200Pro instrument. The data were analysed and fit to a Michaelis–Menton model using OriginPro 2018b (OriginLab Corp, Northampton, MA, USA).

For the PARylation analysis, PARP1 (0.5 μM) and/or nucleosome (0.5 μM NUC167) were incubated with 5 mM NAD+ in a buffer composed of 20 mM Tris (pH 7.5), 100 mM NaCl, 5 mM MgCl_2 and 1 mM DTT. Samples were withdrawn at different time points and quenched by mixing with 5X loading dye (300 mM Tris-HCl [pH 6.8], 50% v/v glycerol, 10% w/v sodium dodecyl sulfate, 0.5% w/v bromophenol blue, 500 mM β-mercaptoethanol). After heat treatment at 95 °C for 1 min, samples were subjected to analysis by 12% sodium dodecyl sulfate-polyacrylamide gel electrophoresis, and gels were stained with Instant Blue (Expedeon, Heidelberg, Germany). The target selectivity and extent of PARylation were apparent from supershifting of the molecular species.

**NCP X-ray crystallographic analysis**. Structures were solved for NCP assembled with canonical (H2A-containing) histone octamer and either a palindromic 145 bp α-satellite[34] or the 147s (sticky-end; described above) DNA fragment. H2A.X-NCP structures were solved for octamer assembled with either the 145 bp α-satellite[34] (H2A.X), the 147s (γH2A.X) or the palindromic 145 bp 601L[33] (H2A.X-S139E) DNA fragment. NCP was crystallized and stabilized in harvest buffer[42], prior to direct mounting into a cryocooling N_2 gas stream set at −175 °C[43]. X-ray diffraction data were recorded at beam line X06DA of the Swiss Light Source (Paul Scherrer Institute, Villigen, Switzerland) using a Pilatus 2M-F detector and an X-ray wavelength of 1.0 Å.

Diffraction data were indexed, integrated, merged, scaled and evaluated with a combination of iMosflm[44], XDS[45], SCALA and AIMLESS from the CCP4 package[46] and the in-house data processing pipeline, go.com, developed by the Swiss Light Source macromolecular cyrstallography beamlines (Paul Scherrer Institute, Villigen, Switzerland)[45,47–49]. Initial phases were obtained by molecular replacement using the program PHASER[50], with previously reported NCP structures assembled with the same or similar DNA fragment serving as the search model. Atomic refinement and model building were carried out with REFMAC[51] and COOT[52], respectively, from the CCP4 suite[46,48]. Graphic figures were prepared with PyMOL (DeLano Scientific LLC, San Carlos, CA, USA).

**Reporting summary**. Further information on research design is available in the Nature Research Reporting Summary linked to this article.

## Data availability
Atomic coordinates and structure factors for the H2A.X-NCP, γH2A.X-NCP, H2A.X-S139E-NCP, H2A-147s-NCP and H2A-(145 bp)-NCP models have been deposited in the Protein Data Bank under accession codes 6K1J, 6K1I, 6K1K, 6JXD and 6IPU, respectively. Data supporting the findings of this work are available within the article and Supplementary Information files. The raw data underlying Figs. 1b, 4 and 5, Supplementary Figs. 3, 6, 7 and 8 and Supplementary Table 1 are provided as a Source Data file. All other data and constructs unique to this study are available from the corresponding author upon reasonable request.

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

## Acknowledgements

We are grateful to V. Olieric, M. Wang, and staff at the Swiss Light Source (Paul Scherrer Institute, Villigen, Switzerland). We thank Y.H. Yau and Y.W. Chuah for assistance with SPR data collection and processing and E. Lai for his kind support on the BLI measurements. Financial sponsorship came from the Singapore Ministry of Education Academic Research Fund Tier 1 (grants 2014-T1-001-049 and 2017-T1-002-020), Tier 2 (grant MOE2015-T2-2-089) and Tier 3 (grant MOE2012-T3-1-001) Programmes. The research leading to these results has received funding from the European Union's Horizon 2020 research and innovation programme under grant agreement #730872, project CALIPSOplus.

## Author contributions

D.S. designed research, conducted experiments and contributed to manuscript writing; L.D.F. carried out X-ray crystallographic work in generating a successful sticky-end NCP DNA (147s) construct, in solving the H2A-147s-NCP and H2A-(145 bp)-NCP structures and in contributing to solving the γH2A.X-147s-NCP structure; S.P. contributed to the crystallographic analysis of H2A.X-NCP and H2A.X-S139E-NCP; C.R. and C.F.L. carried out the synthesis of γH2A.X; S.G.S. provided expertise and guidance for the SPR-based measurements; C.A.D. designed research and wrote the manuscript.

## Competing interests

The authors declare no competing interests.
