## [Peer Review File · Nature Communications]

Reviewers' comments:

Reviewer #1 (Remarks to the Author):

The authors determined three types of crystal structures of the nucleosomes containing H2AX, gammaH2AX, and H2AXS139E. Overall structures of these nucleosomes are just the same as that of the regular nucleosome containing canonical H2A. The crystallography and structural analyses are properly done, but there is no new specific insight to understand function of the H2AX in the nucleosome. Biochemical analyses are not properly performed, and conclusion depicted is misleading.

The nucleosome stability depends on an ensemble of hydrophobic and electrostatic interactions within the complex. High salt decreases the electrostatic interactions, but increase hydrophobic interactions. Therefore, the nucleosome stability should be evaluated by the other methods, such as FRET technology. The present salt-dependent stability data are far from conclusive, and are not reliable.

The authors suggested that the H2AX specific residues are responsible for the instability of the nucleosome with H2AX. To conclude this, mutation analyses with the H2A-type and H2AX-type amino acid replacements must be performed.

Reviewer #2 (Remarks to the Author):

This manuscript describes a biophysical approach to understanding the interaction of PARP1 with the H2A.X nucleosome and its role in recognizing double stranded breaks. Solution physicochemical experiments measuring the kinetics of PARP1-nucleosome interactions reveal that PARP1 binds faster to and has a higher affinity for the H2A.X nucleosome. Concomitantly, the stability of the H2A.X nucleosome is reduced. Crystal structures of the H2A.X and gammaH2A.X nucleosomes are presented and provide a rational basis for the decreased stability. The subject of the paper is timely and important. The experiments are well conceived and executed and the results of high technical quality. The results are discussed appropriately. I have only three minor comments.

In the Abstract and on page 11. I assume what is meant by "histone assemblies" is H3/H4 tetramers, and H2A/H2B dimers. I would explicitly state this.

Bottom of page 6. The phrase, "we see here that this is primarily the resulttowards NCP145 and NUC155" is missing something.

Page 10, beginning of Discussion. The sentence "For example, H2A.Z serves a thermosensory capacity in plants" should be deleted.

Reviewer #3 (Remarks to the Author):

PARP1-mediated poly(ADP-ribosylation) is a key step in the recognition of damaged DNA and its repair. PARP1 specifically binds to both single- and double-stranded DNA breaks, which activates the enzyme, resulting in poly(ADP-ribosylation) of PARP1 itself and of histones and other proteins in the vicinity of the break which, in turn, stimulates the recruitment of DNA repair factors. Genomic regions that are more prone to DNA damage tend to be marked by nucleosomes containing H2A.X, a variant of H2A which has a very different C-terminal tail domain but otherwise differs from H2A at just five residues. These damage-prone regions are also associated with increased levels of PARP1.

Here, the authors address the molecular mechanism by which PARP1 recognises H2A.X-nucleosomes. Using surface plasmon resonance and bilayer interferometry, they show that PARP1 has a ~4-fold higher affinity for H2A.X-nucleosomes versus H2A-nucleosomes. They find that H2A.X-nucleosomes are less stable to salt than H2A-nucleosomes using tyrosine fluorescence spectroscopy. They explain this loss of stability by solving the crystal structure of the H2A.X-nucleosome, showing that altered residues in the H2A.X core domain weaken the interface between H2A and H4 through the loss of a hydrogen bond (H2A-R99 vs. H2A.X G99) and weaken the only contact between the two H2A-H2B dimers (H2A N38 vs. H2A.X H38). The latter leads to a juxtaposition of two histidine residues (H2A.X-H38 and H2B-H79), which would be destabilising if both were protonated.

Another key early step in DNA damage detection is the phosphorylation of H2A.X on serine-139 in the C-tail domain ("gamma-H2A.X"). The authors show that this phosphorylation and a phosphorylation mimic (the S139E mutation in the C-tail of H2A.X) do not affect the binding of PARP1 or nucleosome structure. However, H2A.X phosphorylation does affect the kinetics of poly(ADP-ribosylation), such that the K_m of PARP1 for NAD is lower in the presence of gamma-H2A.X nucleosomes than it is with unphosphorylated H2A.X and H2A-nucleosomes.

These observations suggest a model in which a double-stranded DNA break results in PARP1 binding and subsequent auto-modification, then H2A.X phosphorylation by ATM kinase, followed by retention of PARP1 activity at gamma-H2A.X nucleosomes as cells are depleted of NAD.

This is an intriguing paper with significant insight into the early stages of DNA damage repair. I have the following fairly minor comments:

1. Page 5: "The DSB constructs...". The authors should make clear that the DSBs are represented by the ends of the DNA fragment used to assemble a nucleosome.
2. Page 5, bottom para and Fig. 1b: "Moreover, the selectivity towards nucleosomal over naked DNA.....". Is this correct? Full-length PARP1 has similar affinities for DNA, nucleosome core particles (NCP) and the NUC155 nucleosome; PARP1 does show increased affinity for NUC167. Is this because there is some linker DNA on both sides of the NUC167 nucleosome? Does one PARP1 molecule bind to each DNA end?
3. Fig. 2 shows clearly that H2A.X-nucleosomes are somewhat less stable to NaCl dissociation than H2A-nucleosomes. My understanding of the method is that the tyrosine residues buried at the interfaces between the H3-H4 tetramer and the H2A-H2B dimers within the nucleosome show increased fluorescence as they are exposed to solvent when the H2A-H2B dimers dissociate. On page 10 (bottom para.), the authors argue that this experiment shows reduced nucleosome stability. Is this equivalent to thermodynamic stability?
4. Fig. 5a shows the kinetics of poly(ADP-ribosylation) of the various nucleosomal substrates as a function of NAD concentration. Clearly, PARP1 has a lower K_m for the gamma-H2A.X nucleosome than for the other nucleosomes. Why doesn't the H2A.X-S139E mutant show the same effect, given that it is supposed to be a phospho-mimic?
5. Fig. 5: Which proteins are poly(ADP-ribosylated) in these nucleosome reactions? Do the histones (H2A.X in particular) get modified or is it restricted to PARP1 auto-modification?

RESPONSES TO REVIEWER COMMENTS

We thank the reviewers wholeheartedly for their valuable time and input on our manuscript. We have carefully considered all of the recommendations of the reviewers, whose insightful comments have been very helpful in compiling the revised version. Below we outline the revisions implemented for each of the points raised by the reviewers.

In response to comments by Reviewer #1:

1. *The authors determined three types of crystal structures of the nucleosomes containing H2AX, gammaH2AX, and H2AXS139E. Overall structures of these nucleosomes are just the same as that of the regular nucleosome containing canonical H2A. The crystallography and structural analyses are properly done, but there is no new specific insight to understand function of the H2AX in the nucleosome. Biochemical analyses are not properly performed, and conclusion depicted is misleading.*

The nucleosome stability depends on an ensemble of hydrophobic and electrostatic interactions within the complex. High salt decreases the electrostatic interactions, but increase hydrophobic interactions. Therefore, the nucleosome stability should be evaluated by the other methods, such as FRET technology. The present salt-dependent stability data are far from conclusive, and are not reliable.

The authors suggested that the H2AX specific residues are responsible for the instability of the nucleosome with H2AX. To conclude this, mutation analyses with the H2A-type and H2AX-type amino acid replacements must be performed.

We have conducted additional stability measurements using three different methods. These include, beyond the original salt stability assay, two distinct thermal stability assays and a nuclease sensitivity assay. We have also generated mutants of both H2A and H2A.X, which were assembled into nucleosomes and subjected to stability analysis. The results (including Fig. 4 and Supplementary Fig. 6) and discussion sections have been extensively modified in light of the additional data and insights offered.

In response to comments by Reviewer #2:

This manuscript describes a biophysical approach to understanding the interaction of PARP1 with the H2A.X nucleosome and its role in recognizing double stranded breaks. Solution physicochemical experiments measuring the kinetics of PARP1-nucleosome interactions reveal that PARP1 binds faster to and has a higher affinity for the H2A.X nucleosome. Concomitantly, the stability of the H2A.X nucleosome is reduced. Crystal structures of the H2A.X and gammaH2A.X nucleosomes are presented and provide a rational basis for the decreased stability. The subject of the paper is timely and important. The experiments are well conceived and executed and the results of high technical quality. The results are discussed appropriately. I have only three minor comments.

2. *In the Abstract and on page 11. I assume what is meant by “histone assemblies” is H3/H4 tetramers, and H2A/H2B dimers. I would explicitly state this.*

The wording has been modified in both places to ensure clarity.

3. *Bottom of page 6. The phrase, “we see here that this is primarily the resulttowards NCP145 and NUC155” is missing something.*

The sentence has been split up and reworded for clarity.

4. *Page 10, beginning of Discussion. The sentence “For example, H2A.Z serves a thermosensory capacity in plants” should be deleted.*

The sentence has been deleted.

In response to comments by Reviewer #3:

PARP1-mediated poly(ADP-ribosylation) is a key step in the recognition of damaged DNA and its repair. PARP1 specifically binds to both single- and double-stranded DNA breaks, which activates the enzyme, resulting in poly(ADP-ribosylation) of PARP1 itself and of histones and other proteins in the vicinity of the break which, in turn, stimulates the recruitment of DNA repair factors. Genomic regions that are more prone to DNA damage tend to be marked by nucleosomes containing H2A.X, a variant of H2A which has a very different C-terminal tail domain but otherwise differs from H2A at just five residues. These damage-prone regions are also associated with increased levels of PARP1.

Here, the authors address the molecular mechanism by which PARP1 recognises H2A.X-nucleosomes. Using surface plasmon resonance and biolayer interferometry, they show that PARP1 has a ~4-fold higher affinity for H2A.X-nucleosomes versus H2A-nucleosomes. They find that H2A.X-nucleosomes are less stable to salt than H2A-nucleosomes using tyrosine fluorescence spectroscopy. They explain this loss of stability by solving the crystal structure of the H2A.X-nucleosome, showing that altered residues in the H2A.X core domain weaken the interface between H2A and H4 through the loss of a hydrogen bond (H2A-R99 vs. H2A.X G99) and weaken the only contact between the two H2A-H2B dimers (H2A N38 vs. H2A.X H38). The latter leads to a juxtaposition of two histidine residues (H2A.X-H38 and H2B-H79), which would be destabilising if both were protonated.

Another key early step in DNA damage detection is the phosphorylation of H2A.X on serine-139 in the C-tail domain (“gamma-H2A.X”). The authors show that this phosphorylation and a phosphorylation mimic (the S139E mutation in the C-tail of H2A.X) do not affect the binding of PARP1 or nucleosome structure. However, H2A.X phosphorylation does affect the kinetics of poly(ADP-ribosylation), such that the K_m of PARP1 for NAD is lower in the presence of gamma-H2A.X nucleosomes than it is with unphosphorylated H2A.X and H2A-nucleosomes.

These observations suggest a model in which a double-stranded DNA break results in PARP1 binding and subsequent auto-modification, then H2A.X phosphorylation by ATM kinase, followed by retention of PARP1 activity at gamma-H2A.X nucleosomes as cells are depleted of NAD.

This is an intriguing paper with significant insight into the early stages of DNA damage repair. I have the following fairly minor comments:

5. Page 5: "The DSB constructs...". The authors should make clear that the DSBs are represented by the ends of the DNA fragment used to assemble a nucleosome.

A clarification has been added in this regard.

6. Page 5, bottom para and Fig. 1b: "Moreover, the selectivity towards nucleosomal over naked DNA.....". Is this correct? Full-length PARP1 has similar affinities for DNA, nucleosome core particles (NCP) and the NUC155 nucleosome; PARP1 does show increased affinity for NUC167. Is this because there is some linker DNA on both sides of the NUC167 nucleosome? Does one PARP1 molecule bind to each DNA end?

We have added a clarification in the Results section and also included an elaborated explanation of this aspect in the Introduction (paragraph spanning page 3-4). The study described in reference 6 (Clark et al., 2012) had earlier shed light on this phenomenon.

7. Fig. 2 shows clearly that H2A.X-nucleosomes are somewhat less stable to NaCl dissociation than H2A-nucleosomes. My understanding of the method is that the tyrosine residues buried at the interfaces between the H3-H4 tetramer and the H2A-H2B dimers within the nucleosome show increased fluorescence as they are exposed to solvent when the H2A-H2B dimers dissociate. On page 10 (bottom para.), the authors argue that this experiment shows reduced nucleosome stability. Is this equivalent to thermodynamic stability?

This is reflective of salt stability, strictly speaking. To expand on this issue, we had conducted additional experiments as requested by reviewer #1. Please refer to the corresponding comments.

8. Fig. 5a shows the kinetics of poly(ADP-ribosylation) of the various nucleosomal substrates as a function of NAD concentration. Clearly, PARP1 has a lower K_m for the gamma-H2A.X nucleosome than for the other nucleosomes. Why doesn't the H2A.X-S139E mutant show the same effect, given that it is supposed to be a phospho-mimic?

Indeed this is intriguing and suggests the possibility of specific interaction between the H2A.X tail and PARP1. This has now been mentioned in the Discussion.

9. Fig. 5: Which proteins are poly(ADP-ribosylated) in these nucleosome reactions? Do the histones (H2A.X in particular) get modified or is it restricted to PARP1 auto-modification?

It is only PARP1 that gets PARylated, thus auto-modification. We have included a description of this in the Results, insights in the Discussion and added a figure (Suppl. Figure 8) showing the PARylation analysis.

REVIEWERS' COMMENTS:

Reviewer #1 (Remarks to the Author):

The revised manuscript has been adequately addressed, but information for reproducibility of the MNase assay is still missing. Please make sure the reproducibility for MNase assay, and recommend to add information how many times did the authors repeat this experiment.

Reviewer #3 (Remarks to the Author):

The manuscript is improved. The authors have answered my questions satisfactorily.

Final comment: There should be a DNA size marker in new Figure 4.

RESPONSES TO REVIEWER COMMENTS

We would like to thank all of the reviewers again for their valuable time and input on our manuscript. Below we respond to the final points raised.

In response to comments by Reviewer #1:

1. *The revised manuscript has been adequately addressed, but information for reproducibility of the MNase assay is still missing. Please make sure the reproducibility for MNase assay, and recommend to add information how many times did the authors repeat this experiment.*

We conducted four independent experiments comparing the MNase (micrococcal nuclease) sensitivity of H2A-, H2A.X- and γ H2A.X-nucleosomes, which all show the same characteristics, qualitatively. We have added a description of this in the Methods section.

In response to comments by Reviewer #3:

2. *The manuscript is improved. The authors have answered my questions satisfactorily. Final comment: There should be a DNA size marker in new Figure 4.*

We were not able to achieve sufficient resolution of the individual DNA fragments in digest profiles when using smaller comb sizes (giving more than 15 wells). We therefore show a single gel, but with all of the 15 H2A, H2A.X and γ H2A.X samples together and provide an additional gel image in the Supplementary Information, which has a 10 bp DNA ladder in conjunction with the digest profile. We have added a comment in the Figure 4 legend in reference to Supplementary Fig. 6, which has the DNA marker.